# Further Geometric Properties of the Barnes–Mittag-Leffler Function

**Abdulaziz Alenazi** [1],[†] and **Khaled Mehrez** [2],[*],[†]

[1] Department of Mathematics, College of Science, Northern Border University, Arar 73213, Saudi Arabia; a.alenazi@nbu.edu.sa
[2] Department of Mathematics, IPEIK Kairouan, University of Kairouan, Kairouan 3100, Tunisia
[*] Correspondence: k.mehrez@yahoo.fr
[†] These authors contributed equally to this work.

**Abstract:** In this paper, we find sufficient conditions to be imposed on the parameters of a class of functions related to the Barnes–Mittag-Leffler function that allow us to conclude that it possesses certain geometric properties (such as starlikeness, uniformly starlike (convex), strongly starlike (convex), convexity, and close-to-convexity) in the unit disk. The key tools in some of our proofs are the monotonicity properties of a certain class of functions related to the gamma function.

**Keywords:** Barnes–Mittag-Leffler function; analytic function; univalent function; starlike function; convex function

**MSC:** 30D15; 30C45; 30H10

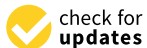



## 1. Introduction and Motivation

The study of the geometric properties of some classes of analytic functions associated with some special functions in the unit disk in the complex plane has always attracted several researchers. One of the special functions for which the geometric properties have been studied widely is the Mittag-Leffler function [1–4]. Interested readers can find more information on the various geometric properties of certain analytic functions like the Wright function [5], generalized Bessel function [6–8], and Fox–Wright functions [9] in the listed references. For the geometric behavior of other special functions, one can refer to [10–15] and the references cited therein.

The Mittag-Leffler function is closely related to the Barnes–Mittag-Leffler function. The Mittag-Leffler function play a crucial role in fractional calculus, approximation theory, and various branches of science and engineering. Our main goal of the present paper is to study several potentially geometric properties of the normalized form of the Barnes–Mittag-Leffler function. This paper is a continuation along some lines of the authors' previous results.

Now, we recall some known definitions and results related to the context of Geometric Functions Theory. Let $\mathcal{H}$ denote the class of all analytic functions inside the unit disk

$$\mathcal{D} = \left\{ z : z \in \mathbb{C} \ \text{and} \ |z| < 1 \right\}.$$

Assume that $\mathcal{A}$ denotes the collection of all functions $\varphi \in \mathcal{H}$, satisfying the normalization $\varphi(0) = \varphi'(0) - 1 = 0$ such that

$$\varphi(z) = z + \sum_{k=2}^{\infty} a_k z^k \quad (\forall z \in \mathcal{D}). \tag{1}$$

A function $\varphi \in \mathcal{A}$ is said to be a starlike function (with respect to the origin 0) in $\mathcal{D}$ if $\varphi$ is univalent in $\mathcal{D}$ and $\varphi(\mathcal{D})$ is a starlike domain with respect to 0 in $\mathbb{C}$. The analytic characterization of the class of starlike functions is given below [16]:

$$\Re\left(\frac{z\varphi'(z)}{\varphi(z)}\right) > 0 \quad (\forall z \in \mathcal{D}).$$

Some geometric characterization of $k$-starlike functions is given in [17] and the references therein.

If $\varphi(z)$ is a univalent function in $\mathcal{D}$ and $\varphi(\mathcal{D})$ is a convex domain in $\mathbb{C}$, then $\varphi \in \mathcal{A}$ is said to be a convex function in $\mathcal{D}$. The class of convex functions can be described as follows:

$$\Re\left(1 + \frac{z\varphi''(z)}{\varphi'(z)}\right) > 0 \quad (\forall z \in \mathcal{D}).$$

However, an analytic function $\varphi$ is convex if and only if the function $z\varphi'$ is starlike.

An analytic function $\varphi$ in $\mathcal{A}$ is called close-to-convex in the open unit disk $\mathcal{D}$ if there exists a function $\phi(z)$ that is starlike in $\mathcal{D}$ such that

$$\Re\left(\frac{z\varphi'(z)}{\varphi(z)}\right) > 0 \quad (\forall z \in \mathcal{D}).$$

It can be easily verified that every starlike (and hence, convex) function is close-to-convex. It can be noted that every close-to-convex function in $\mathcal{D}$ is also univalent in $\mathcal{D}$.

A function $\varphi$ in $\mathcal{A}$ is called uniformly convex (or uniformly starlike) in $\mathcal{D}$ if for every circular arc $\xi$ contained in $\mathcal{D}$ with center $\kappa \in \mathcal{D}$, the image arc $\varphi(\xi)$ is convex or starlike with respect to $\varphi(\kappa)$; for more details, see [18]. This class of functions is denoted by UCV; the analytic description of the class of uniformly convex function is given as [19]:

$$\left|\frac{z\varphi^{(2)}(z)}{\varphi'(z)}\right| < \frac{1}{2} \ \Rightarrow \ \varphi \in UCV \quad (\forall \ \varphi \in \mathcal{A}). \tag{2}$$

An analytic function $\varphi \in \mathcal{A}$ is said to be strongly starlike of order $\theta$ if and only if

$$\left|\arg\left(\frac{z\varphi'(z)}{\varphi(z)}\right)\right| < \frac{\pi}{2}\theta \quad (\forall z \in \mathcal{D}),$$

where $0 < \theta \leq 1$. We denote that if $\theta = 1$, then $\varphi$ is starlike (or strongly starlike of order 1).

An analytic function $\varphi \in \mathcal{A}$ is said to be strongly convex of order $\eta$ if and only if

$$\left|\arg\left(\frac{(z\varphi'(z))'}{\varphi'(z)}\right)\right| < \frac{\pi}{2}\eta \quad (\forall z \in \mathcal{D}),$$

where $0 < \eta \leq 1$. We denote that if $\theta = 1$, then $\varphi$ is convex (or strongly convex of order 1) in $\mathcal{D}$.

Let $\varphi_1$ and $\varphi_2$ be two analytic functions in $\mathcal{D}$. The function $\varphi_1$ is said to be subordinate to the function $\varphi_2$, denoted by

$$\varphi_1 \prec \varphi_2 \quad (\varphi_1(z) \prec \varphi_2(z) \ \ \forall z \in \mathcal{D}),$$

if $\varphi_1 = \varphi_2 \circ \phi$, where $\phi : \mathcal{D} \to \mathcal{D}$ is an analytic function in $\mathcal{D}$ such that $\phi(0) = 0$. Hence, in view of the Lemma of Schwartz, we deduce that $\varphi_1 \prec \varphi_2$ if and only if $\varphi_1(0) = \varphi_2(0)$ and $\varphi_1(\mathcal{D}) \subseteq \varphi_2(\mathcal{D})$.

Now, we introduce the definitions of the Mittag-Leffler function $E_\kappa(z)$ and its two-parameter version $E_{\kappa,\nu}(z)$, respectively, as defined by [20,21]:

$$E_\kappa(z) = \sum_{n=0}^{\infty} \frac{z^n}{\Gamma(\kappa n + 1)} \quad (z \in \mathbb{C}, \ \kappa \geq 0), \tag{3}$$

and

$$E_{\kappa,\nu}(z) = \sum_{n=0}^{\infty} \frac{z^n}{\Gamma(\kappa n + \nu)} \quad (z, \kappa, \nu \in \mathbb{C}, \ \Re(\kappa) > 0, \Re(\nu) > 0). \tag{4}$$

For some of the properties of the Mittag-Leffler function, we refer the reader to [22,23] and the references cited therein.

One of the most important generalizations of the Mittag-Leffler function is the Barnes–Mittag-Leffler function $E_{\kappa,\nu}^{(a)}(s;z)$, which is defined [24] as

$$E_{\kappa,\nu}^{(a)}(s;z) := \sum_{n=0}^{\infty} \frac{z^n}{\Gamma(\kappa n + \nu)(n + a)^s}, \quad (a, \nu \in \mathbb{C} \setminus \mathbb{Z}_0^-, \Re(\kappa) > 0, s, z \in \mathbb{C}). \tag{5}$$

It can be noted that $E_{\kappa,\nu}^{(a)}(0;z) = E_{\kappa,\nu}(z)$, $E_{\kappa,1}(z) = E_\kappa(z)$, and $E_1(z) = \exp(z)$.

By using the fact that $E_{\kappa,\nu}^{(a)}(s;z) \notin \mathcal{A}$, we introduce the following normalization of the Barnes–Mittag-Leffler function:

$$\begin{aligned}
\mathcal{E}_{\kappa,\nu}^{(a)}(s;z) &= a^s \Gamma(\nu) z E_{\kappa,\nu}^{(a)}(s;z) \\
&= \sum_{n=1}^{\infty} \rho_n^{(a)}(\nu, \kappa, s) z^k,
\end{aligned} \tag{6}$$

where

$$\rho_n^{(a)}(\nu, \kappa, s) = \frac{a^s \Gamma(\nu)}{\Gamma(n\kappa + \nu - \kappa)(n + a - 1)^s}, \ n \geq 1.$$

In the present paper, we will restrict our attention to the conditions involving positive real-valued parameters $a, s, \nu$, and $\kappa$ and the argument $z \in \mathbb{C}$. In this paper, we study some geometric properties (such as starlikness, uniformly starlike (convex), strongly starlike (convex), convexity, close-to-convexity) of a class of analytic function related to the Barnes–Mittag-Leffler function (consult (8)).

At the end of this section, we offer some helpful lemmas that aid with the completion of the proofs of the major findings.

**Lemma 1** ([25]). *Assume that $f \in \mathcal{A}$. If the inequality*

$$\left| \frac{f(z)}{z} - 1 \right| < 1$$

*holds for all $z \in \mathcal{D}$, then $f$ is starlike in*

$$\mathcal{D}_{\frac{1}{2}} := \left\{ z \in \mathbb{C} \text{ and } |z| < \frac{1}{2} \right\}.$$

**Lemma 2** ([26]). *Assume that $f \in \mathcal{A}$ and $|f'(z) - 1| < 1$ is satisfied for each $z \in \mathcal{D}$; then $f$ is convex in $\mathcal{D}_{\frac{1}{2}}$.*

**Lemma 3** ([27]). *If $F$ such that $F(0) = 1$ is convex univalent in $\mathcal{D}$ and $G$ with $G(0) = 1$ is analytic in $\mathcal{D}$ such that $G \prec F$ in $\mathcal{D}$, then we get*

$$(m+1)z^{-1-m}\int_0^z s^m G(s)ds \prec (m+1)z^{-1-m}\int_0^z s^m F(s)ds, \quad \forall n \in \mathbb{N} \cup \{0\}.$$

**Lemma 4** (Ozaki [28]). *If $\varphi$ is of the form* (1) *such that*

$$1 \leq 2a_2 \leq \ldots \leq \ell a_\ell \leq (\ell+1)a_{\ell+1} \leq \ldots \leq 2,$$

*or if*

$$1 \geq 2a_2 \geq \ldots \geq \ell a_\ell \geq (\ell+1)a_{\ell+1} \geq \ldots \geq 0,$$

*then $\varphi$ is close-to-convex with respect to the function $-\log(1 - z)$.*

**Lemma 5** ([29]). *Assume that the real sequence $(x_n)_{n\geq 1}$ is positive and decreasing and satisfies $x_1 = 1$. If $(x_n)_{n\geq 1}$ is a convex sequence, then*

$$\Re\left(\sum_{n=1}^\infty x_n z^{n-1}\right) > \frac{1}{2} \quad (\forall z \in \mathcal{D}).$$

## 2. Main Results and Their Consequences

Our first major finding is asserted by the following theorem.

**Theorem 1.** *Assume that one of the following sets of conditions holds true:*
$(H_1)$ : *Suppose that $\nu + \kappa > x^*, \Gamma(\nu) \leq \Gamma(\nu + \kappa)$, and $a^s \zeta(s) \leq 1$   $(s > 1)$, where $x^* \approx 1.461632144\cdots$ is the abscissa of the minimum of the gamma function, and $\zeta(s)$ is the Riemann zeta function defined by*

$$\zeta(s) = \sum_{n=1}^\infty \frac{1}{n^s} \quad (\Re(s) > 1).$$

$(H_2)$ : *Suppose that $\Gamma(\nu)(e - 1) \leq \Gamma(\nu + \kappa)$, and also, the following inequality*

$$\kappa \log(\kappa + \nu) - \frac{\kappa}{\kappa + \nu} \geq \log(2) - \frac{1}{4} \tag{7}$$

*is valid. Then, the function $\mathcal{E}_{\kappa,\nu}^{(a)}(s;z)$ is starlike in $\mathcal{D}_{\frac{1}{2}}$.*

**Proof.** Let $z \in \mathcal{D}$, and assume that the conditions of $(H_1)$ are valid; then we have

$$\left|\frac{\mathcal{E}_{\kappa,\nu}^{(a)}(s;z)}{z} - 1\right| < \sum_{n=2}^\infty \rho_n^{(a)}(\nu, \kappa, s). \tag{8}$$

Since $\nu + \kappa > x^*$, and using the fact that the Gamma function $z \mapsto \Gamma(z)$ is increasing on $(x^*, \infty)$, we deduce that

$$\Gamma(\nu + \kappa) \leq \Gamma(n\kappa + \nu) \quad (\forall n \geq 2).$$

Hence, in view of the above inequality and (8) combined with the condition $\Gamma(\nu) \leq \Gamma(\nu + \kappa)$, we obtain

$$\left|\frac{\mathcal{E}_{\kappa,\nu}^{(a)}(s;z)}{z} - 1\right| < \sum_{n=1}^\infty \left(\frac{a}{n+a}\right)^s$$
$$< a^s \zeta(s)$$
$$\leq 1. \tag{9}$$

Then, with the aid of Lemma 1, we conclude that the function $\mathcal{E}_{\kappa,\nu}^{(a)}(s;z)$ is starlike in $\mathcal{D}_{\frac{1}{2}}$. Finally, we assume that the hypotheses $(H_2)$ hold true; then we have

$$\left| \frac{\mathcal{E}_{\kappa,\nu}^{(a)}(s;z)}{z} - 1 \right| < \sum_{n=1}^{\infty} \frac{\tilde{\rho}_{n+1}^{(a)}(\nu,\kappa,s)}{n!}, \tag{10}$$

where

$$\hat{\rho}_{n+1}^{(a)}(\nu,\kappa,s) = n! \rho_{n+1}^{(a)}(\nu,\kappa,s).$$

In [30], Proof of Theorem 2.8, the authors proved that the function

$$g_1(t) = \frac{\Gamma(t+1)}{\Gamma(\kappa t + \nu)}$$

is decreasing on $(1,\infty)$ if the inequality (7) holds. By these observations and under the conditions of $(H_2)$, we deduce that

$$\begin{aligned} \left| \frac{\mathcal{E}_{\kappa,\nu}^{(a)}(s;z)}{z} - 1 \right| &< \frac{\Gamma(\nu)}{\Gamma(\nu+\kappa)} \sum_{n=1}^{\infty} \frac{a^s}{n!(n+a)^s} \\ &< \frac{\Gamma(\nu)}{\Gamma(\nu+\kappa)} \sum_{n=1}^{\infty} \frac{1}{n!} \\ &= \frac{(e-1)\Gamma(\nu)}{\Gamma(\nu+\kappa)} \\ &\leq 1. \end{aligned} \tag{11}$$

Again, by means of Lemma 1, we derive the desired result asserted by Theorem 1 under the hypotheses $(H_2)$. □

**Corollary 1.** *Assume that $\nu + \kappa > x^*, \Gamma(\nu) \leq \Gamma(\nu+\kappa)$, and $\sqrt{6}a \leq \pi$; then the function $\mathcal{E}_{\kappa,\nu}^{(a)}(2;z)$ is starlike in $\mathcal{D}_{\frac{1}{2}}$.*

**Proof.** Setting $s = 2$ in the hypotheses $(H_1)$ of Theorem 1, we obtain the required result. □

**Example 1.** *The function $\mathcal{E}_{\frac{1}{2},\frac{3}{2}}^{(1)}(2;z)$ is starlike in $\mathcal{D}_{\frac{1}{2}}$.*

**Remark 1.** *If we set $s = 0$ in Theorem 1 under the conditions $(H_2)$, we obtain that the normalized form of the Mittag-Leffler function $\mathcal{E}_{\kappa,\nu}(z)$ defined by*

$$\mathcal{E}_{\kappa,\nu}(z) = \Gamma(\nu)z E_{\kappa,\nu}(z), \tag{12}$$

*is starlike in $\mathcal{D}_{\frac{1}{2}}$.*

**Remark 2.** *In [4], Theorem 2.4, the authors obtained that the function $\mathcal{E}_{\kappa,\nu}(z)$ is starlike in $\mathcal{D}_{\frac{1}{2}}$ for $\kappa \geq 1$ and $\nu \geq \frac{\sqrt{5}+1}{2} \approx 1.61 \cdots$. In view of Remark 1, we see that the function $\mathcal{E}_{\frac{5}{2},\frac{2}{3}}(z)$ is starlike. However, Theorem 1 improves the corresponding results derived in [4].*

**Theorem 2.** *Let the parameters $\kappa$ and $\nu$ satisfy the following inequality:*

$$\kappa \log(\kappa+\nu) - \frac{\kappa}{\kappa+\nu} \geq \log(3) - \frac{1}{6}. \tag{13}$$

*If $2(e-1)\Gamma(\nu) \leq \Gamma(\nu+\kappa)$ or if $2\Gamma(\nu) \leq a^s \zeta(s)\Gamma(\nu+\kappa) \ (s > 1)$, then the function $\mathcal{E}_{\kappa,\nu}^{(a)}(s;z)$ is convex in $\mathcal{D}_{\frac{1}{2}}$.*

**Proof.** For $z \in \mathcal{D}$, we have

$$\left| \left( \mathcal{E}_{\kappa,\nu}^{(a)}(s;z) \right)' - 1 \right| < \sum_{n=1}^{\infty} (n+1) \rho_{n+1}^{(a)}(\nu,\kappa,s)$$
$$= \sum_{n=1}^{\infty} \frac{\tilde{\rho}_{n+1}^{(a)}(\nu,\kappa,s)}{n!}, \tag{14}$$

where

$$\tilde{\rho}_{n+1}^{(a)}(\nu,\kappa,s) = (n+1)! \rho_{n+1}^{(a)}(\nu,\kappa,s) \quad (n \geq 1). \tag{15}$$

However, in view of the fact that the function (cf. [30], Proof of Theorem 3.8)

$$t \mapsto \frac{\Gamma(t+2)}{\Gamma(\kappa t + \nu)}$$

is decreasing on $(1, \infty)$ when the parameters $\kappa$ and $\nu$ satisfy inequality (13) and with the help of (14), we find that

$$\left| \left( \mathcal{E}_{\kappa,\nu}^{(a)}(s;z) \right)' - 1 \right| < \frac{\Gamma(3)\Gamma(\nu)}{\Gamma(\kappa+\nu)} \sum_{n=1}^{\infty} \frac{a^s}{n!(n+a)^s}$$
$$=: \frac{\Gamma(3)\Gamma(\nu)}{\Gamma(\kappa+\nu)} \mathcal{I}(a,s). \tag{16}$$

It is easy to obtain that the following inequalities of the function $\mathcal{I}(a,s)$ read as follows:

$$\mathcal{I}(a,s) \leq (e-1), \tag{17}$$

and

$$\mathcal{I}(a,s) \leq a^s \zeta(s). \tag{18}$$

However, combining the above inequalities with (16), we readily derived the desired result by means of Lemma 2. $\quad \square$

**Corollary 2.** *If $a^s \zeta(s) \geq 1$, then the function $\mathcal{E}_{2,1}^{(a)}(s;z)$ is convex in $\mathcal{D}_{\frac{1}{2}}$. Furthermore, if $a \geq \frac{\sqrt{6}}{\pi}$, then the function $\mathcal{E}_{2,1}^{(a)}(2;z)$ is convex in $\mathcal{D}_{\frac{1}{2}}$.*

**Proof.** Firstly, specifying $\kappa = 2$ and $\nu = 1$ in Theorem 2, we obtained the first stated result asserted by the above corollary. Secondly, taking $s = \kappa = 2$ and $\nu = 1$ in the above theorem, we readily derived the second result in Corollary 2. $\quad \square$

**Remark 3.** *If we set $s = 0$ in the first set of conditions in Theorem 2, we deduce that the normalized form of the Mittag-Leffler function $\mathcal{E}_{\kappa,\nu}(z)$ defined in (12) is convex in $\mathcal{D}_{\frac{1}{2}}$.*

**Remark 4.** *In [4], Theorem 2.4, Bansal and Prajapat proved that the function $\mathbb{E}_{\kappa,\nu}(z)$ is convex in $\mathcal{D}_{\frac{1}{2}}$ if $\kappa \geq 1$ and $\nu \geq \frac{\sqrt{17}+3}{2} \approx 3.65 \cdots$. However, in view of the above remark, we can easily conclude that the function $\mathcal{E}_{\frac{3}{2},\frac{5}{2}}(z)$ is convex in $\mathcal{D}_{\frac{1}{2}}$. Hence, Theorem 2 improves Theorem 2.4 in [4].*

**Theorem 3.** *Let the parameter space be the same as in Theorem 2. Upon setting*

$$\epsilon_1 := \frac{2\Gamma(\nu)(e-1)}{\Gamma(\kappa+\nu)}, \quad \epsilon_2 := \frac{2\Gamma(\nu)a^s \zeta(s)}{\Gamma(\kappa+\nu)} \ (s > 1), \quad \epsilon^* := \min(\epsilon_1, \epsilon_2)$$

*and*

$$\theta = \frac{2}{\pi} \arcsin\left( \frac{\epsilon^*}{2} \left[ \sqrt{4 - (\epsilon^*)^2} + \sqrt{1 - (\epsilon^*)^2} \right] \right),$$

*then the function $\mathcal{E}_{\kappa,\nu}^{(a)}(s;z)$ is strongly starlike of order $\theta$.*

**Proof.** According to (16) and (17) (respectively, (16) and (18)), we get

$$\left| \left( \mathcal{E}_{\kappa,\nu}^{(a)}(s;z) \right)' - 1 \right| < \epsilon^*,$$

where $0 < \epsilon^* \leq 1$. Then, by using the above inequality, we obtain

$$\left( \mathcal{E}_{\kappa,\nu}^{(a)}(s;z) \right)' \prec 1 + \epsilon^* z \quad (\forall z \in \mathcal{D}).$$

This implies that

$$\left| \arg\left( \left( \mathcal{E}_{\kappa,\nu}^{(a)}(s;z) \right)' \right) \right| < \arcsin(\epsilon^*). \tag{19}$$

Now, we apply Lemma 3 for $k = 0$, where $G(z) = \left( \mathcal{E}_{\kappa,\nu}^{(a)}(s;z) \right)'$ and $F(z) = 1 + \epsilon^* z$; we obtain

$$\frac{\mathcal{E}_{\kappa,\nu}^{(a)}(s;z)}{z} \prec 1 + \frac{\epsilon^*}{2} z \quad (\forall z \in \mathcal{D}).$$

Therefore, we get

$$\left| \arg\left( \frac{\mathcal{E}_{\kappa,\nu}^{(a)}(s;z)}{z} \right) \right| < \arcsin\left( \frac{\epsilon^*}{2} \right). \tag{20}$$

However, keeping (19) and (20) in mind, we have

$$\left| \arg\left( \frac{z(\mathcal{E}_{\kappa,\nu}^{(a)}(s;z))'}{\mathcal{E}_{\kappa,\nu}^{(a)}(s;z)} \right) \right| = \left| \arg\left( \left( \mathcal{E}_{\kappa,\nu}^{(a)}(s;z) \right)' \right) - \arg\left( \frac{\mathcal{E}_{\kappa,\nu}^{(a)}(s;z)}{z} \right) \right|$$

$$\leq \left| \arg\left( \left( \mathcal{E}_{\kappa,\nu}^{(a)}(s;z) \right)' \right) \right| + \left| \arg\left( \frac{\mathcal{E}_{\kappa,\nu}^{(a)}(s;z)}{z} \right) \right|$$

$$= \arcsin(\epsilon^*) + \arcsin(\frac{\epsilon^*}{2})$$

$$= \arcsin\left( \frac{\epsilon^*}{2} \left[ \sqrt{4 - (\epsilon^*)^2} + \sqrt{1 - (\epsilon^*)^2} \right] \right)$$

$$= \frac{\theta \pi}{2},$$

and this completes the proof. $\quad\square$

**Remark 5.** *If $\theta = 1$, then the function $\mathcal{E}_{\kappa,\nu}^{(a)}(s;z)$ is starlike in $\mathcal{D}$.*

**Theorem 4.** *Assume that $\min(\nu, \kappa) \geq 1$ and $s > 1$. If $\Gamma(\nu + \kappa) \geq 2\Gamma(\nu)a^s\zeta(s)$, then the function $\mathcal{E}_{\kappa,\nu}^{(a)}(s;z)$ is starlike in $\mathcal{D}$.*

**Proof.** According to the analytic description of starlike functions, to show that the function $\mathcal{E}_{\kappa,\nu}^{(a)}(s;z)$ is starlike in $\mathcal{D}$, it suffices to prove that the following inequality

$$\Re\left( \frac{z(\mathcal{E}_{\kappa,\nu}^{(a)}(s;z))'}{\mathcal{E}_{\kappa,\nu}^{(a)}(s;z)} \right) > 0 \quad (\forall z \in \mathcal{D}).$$

It suffices to establish that the following inequality

$$\left| \frac{z(\mathcal{E}_{\kappa,\nu}^{(a)}(s;z))'}{\mathcal{E}_{\kappa,\nu}^{(a)}(s;z)} - 1 \right| < 1, \tag{21}$$

holds for all $z \in \mathcal{D}$. In view of (8) and by using routine algebra, we have

$$
\begin{aligned}
\left| (\mathcal{E}_{\kappa,\nu}^{(a)}(s;z))' - \frac{\mathcal{E}_{\kappa,\nu}^{(a)}(s;z)}{z} \right| &< \left| \sum_{n=1}^{\infty} n \rho_{n+1}^{(a)}(\nu,\kappa,s) \right| \\
&= \left| \sum_{n=1}^{\infty} \frac{\tilde{\rho}_n^{(a)}(\nu,\kappa,s)}{(n-1)!} \right| \\
&\leq \left| \sum_{n=1}^{\infty} \tilde{\rho}_n^{(a)}(\nu,\kappa,s) \right|,
\end{aligned}
\tag{22}
$$

where $\tilde{\rho}_n^{(a)}(\nu,\kappa,s)$ is defined in (15). Since the digamma function $\psi(z) = \frac{\Gamma'(z)}{\Gamma(z)}$ is increasing on $(0,\infty)$, we deduce that the sequence $(n!/\Gamma(n\kappa + \nu))_{n \geq 1}$ is decreasing for all $\min(\kappa,\nu) \geq 1$. This, in turn, implies

$$
\tilde{\rho}_n^{(a)}(\nu,\kappa,s) \leq \frac{\Gamma(\nu)a^s}{(n+a)^s \Gamma(\kappa+\nu)}.
\tag{23}
$$

Having (22) and (23) in mind, we obtain

$$
\left| (\mathcal{E}_{\kappa,\nu}^{(a)}(s;z))' - \frac{\mathcal{E}_{\kappa,\nu}^{(a)}(s;z)}{z} \right| < \frac{\Gamma(\nu)a^s \zeta(s)}{\Gamma(\nu+\kappa)} \quad (\forall z \in \mathcal{D}).
\tag{24}
$$

Since $\Gamma(n\kappa + \nu) \geq \Gamma(\kappa + \nu)$ for all $\min(n,\nu,\kappa) \geq 1$, for $z \in \mathcal{D}$, we get

$$
\left| \frac{\mathcal{E}_{\kappa,\nu}^{(a)}(s;z)}{z} \right| > 1 - \frac{a^s \zeta(s) \Gamma(\nu)}{\Gamma(\nu+\kappa)} > 0.
\tag{25}
$$

So by combining (24) and (25), we obtain that the inequality (21) is valid under our assumption. This is what we intended to show. $\square$

**Corollary 3.** *Let $s > 1$. If $a^s \zeta(s) \leq \frac{1}{2}$, then the function $\mathcal{E}_{1,1}^{(a)}(s;z)$ is starlike in $\mathcal{D}$. In particular, if $0 < a \leq \frac{\sqrt{3}}{\pi}$, then the function $\mathcal{E}_{1,1}^{(a)}(2;z)$ is starlike in $\mathcal{D}$.*

**Proof.** Firstly, we set $\nu = \kappa = 1$, and secondly, we let $\nu = \kappa = 1$ and $s = 2$ in the above theorem; we derive the desired results asserted by Corollary 3. $\square$

**Theorem 5.** *Suppose that $4a^s \Gamma(\nu)\zeta(s) \leq \Gamma(\nu+\kappa)$. Also, if the following inequality is valid:*

$$
\kappa \log(\nu+\kappa) - \frac{\kappa}{\nu+\kappa} \geq \log(3) + \frac{5}{6},
\tag{26}
$$

*then the function $\mathcal{E}_{\kappa,\nu}^{(a)}(s;z)$ is convex in $\mathcal{D}$.*

**Proof.** According to the analytic characterizations of a convex function, to show that the function $\mathcal{E}_{\kappa,\nu}^{(a)}(s;z)$ is convex in $\mathcal{D}$, it is enough to prove that the function

$$
\mathcal{F}_{\kappa,\nu}^{(a)}(s;z) = z(\mathcal{E}_{\kappa,\nu}^{(a)}(s;z))'
$$

is starlike in $\mathcal{D}$. For this, it suffices to prove the following inequality:

$$
\left| \frac{z(\mathcal{F}_{\kappa,\nu}^{(a)}(s;z))'}{\mathcal{F}_{\kappa,\nu}^{(a)}(s;z)} - 1 \right| < 1 \quad (\forall z \in \mathcal{D}).
\tag{27}
$$

Again, by (8), we have

$$\left| (\mathcal{F}_{\kappa,\nu}^{(a)}(s;z))' - \frac{\mathcal{F}_{\kappa,\nu}^{(a)}(s;z)}{z} \right| < \sum_{n=1}^{\infty} n(n+1)\rho_{n+1}^{(a)}(\nu,\kappa,s)$$

$$= \sum_{n=1}^{\infty} \frac{n\Gamma(n+2)\rho_{n+1}^{(a)}(\nu,\kappa,s)}{n!}. \tag{28}$$

Moreover, in the proof of Theorem 3.1 [30], under the relation (26), the authors proved that the sequence $(\xi_n)_{n\geq 1}$ defined by

$$\xi_n = \frac{n\Gamma(n+2)}{\Gamma(\kappa n+\nu)} \quad (n \geq 1), \tag{29}$$

is decreasing, and consequently, for all $z \in \mathcal{D}$, we obtain

$$\left| (\mathcal{F}_{\kappa,\nu}^{(a)}(s;z))' - \frac{\mathcal{F}_{\kappa,\nu}^{(a)}(s;z)}{z} \right| < \frac{2a^s\zeta(s)\Gamma(\nu)}{\Gamma(\nu+\kappa)}. \tag{30}$$

Elementary calculus gives

$$\left| \frac{\mathcal{F}_{\kappa,\nu}^{(a)}(s;z)}{z} \right| > 1 - \sum_{n=1}^{\infty} (n+1)\rho_{n+1}^{(a)}(\nu,\kappa,s)$$

$$= 1 - \sum_{n=1}^{\infty} \frac{\Gamma(\nu)a^s\chi_n^{(a)}(\nu,\kappa,s)}{n!(n+a)^s}, \tag{31}$$

where $\chi_n^{(a)}(\nu,\kappa,s) = \frac{\xi_n^{(a)}(\nu,\kappa,s)}{n}$ $(n \geq 1)$, and $z \in \mathcal{D}$. Under our assumption, the sequence $(\xi_n)_{n\geq 1}$ is decreasing, and consequently, the sequence $(\chi_n)_{n\geq 1}$ is decreasing. This means that

$$\left| \frac{\mathcal{F}_{\kappa,\nu}^{(a)}(s;z)}{z} \right| \geq 1 - \sum_{n=1}^{\infty} \frac{\Gamma(\nu)a^s\chi_1}{n!(n+a)^s}$$

$$= 1 - \frac{2\Gamma(\nu)a^s}{\Gamma(\nu+\kappa)} \sum_{n=1}^{\infty} \frac{1}{n!(n+a)^s}$$

$$\geq 1 - \frac{2\Gamma(\nu)a^s}{\Gamma(\nu+\kappa)} \sum_{n=1}^{\infty} \frac{1}{(n+a)^s} \tag{32}$$

$$\geq 1 - \frac{2\Gamma(\nu)a^s\zeta(s)}{\Gamma(\nu+\kappa)} > 0.$$

Finally, by combining (30) and (32), we deduce that the inequality (27) holds true under the conditions we imposed on the parameters. With this the proof, Theorem 5 is complete. □

If we set $\nu = \kappa = 2$ in Theorem 5, we compute the following result:

**Corollary 4.** *If $2a^s\zeta(s) \leq 3$, then the function $\mathcal{E}_{2,2}^{(a)}(s;z)$ is convex in $\mathcal{D}$.*

If we let $s = 2$ in Theorem 5, we derive the following result:

**Corollary 5.** *If $0 < a \leq \frac{3}{\pi}$, then the function $\mathcal{E}_{2,2}^{(a)}(2;z)$ is convex in $\mathcal{D}$.*

**Example 2.** *The functions $\mathcal{E}_{2,2}^{(\frac{1}{2})}(2;z)$ and $\mathcal{E}_{2,2}^{(\frac{3}{4})}(2;z)$ are convex in $\mathcal{D}$.*

**Theorem 6.** *Assume that the inequality (26) holds true. Also, suppose that*

$$6a^s\Gamma(\nu)\zeta(s) \leq \Gamma(\nu + \kappa).$$

*Then the function $\mathcal{E}_{\kappa,\nu}^{(a)}(s;z)$ is uniformly convex in $\mathcal{D}$.*

**Proof.** Let $z \in \mathcal{D}$; then by (8), we get

$$z(\mathcal{E}_{\kappa,\nu}^{(a)}(s;z))'' = \sum_{n=1}^{\infty} \xi_n z^n \qquad (33)$$

where $(\xi_n)_{n\geq 1}$ is defined in (29). By using the fact that the sequence $(\xi_n)_{n\geq 1}$ is decreasing, we obtain

$$\left| z(\mathcal{E}_{\kappa,\nu}^{(a)}(s;z))'' \right| \leq \frac{2a^s\zeta(s)\Gamma(\nu)}{\Gamma(\nu + \kappa)}. \qquad (34)$$

Further, for $z \in \mathcal{D}$, we obtain

$$\left| (\mathcal{E}_{\kappa,\nu}^{(a)}(s;z))' \right| \geq 1 - \sum_{n=1}^{\infty} \phi_n^{(a)}(\nu, \kappa, s) \qquad (35)$$

where

$$\phi_n^{(a)}(\nu, \kappa, s) = \left( \frac{n+1}{n} \right) \frac{\xi_n^{(a)}(\nu, \kappa, s)}{\Gamma(n+2)} \quad (n \geq 1).$$

Hence, the sequence $(\phi_n)_{n\geq 1}$ is decreasing as the product of three decreasing sequences. This, in turn, implies that the following inequality holds true:

$$\left| (\mathcal{E}_{\kappa,\nu}^{(a)}(s;z))' \right| \geq \frac{\Gamma(\nu + \kappa) - 2a^s\Gamma(\nu)\zeta(s)}{\Gamma(\nu + \kappa)}.$$

Finally, in virtue of the above inequality and (34) and with the help of the analytic description given in (2), we obtain the desired result. $\square$

Taking $(\nu, \kappa) = (2, 2)$ in Theorem 6, we compute the following result:

**Corollary 6.** *If $a^s\zeta(s) \leq 1$, then the function $\mathcal{E}_{2,2}^{(a)}(s;z)$ is uniformly convex in $\mathcal{D}$.*

Next, we set $s = 2$ in the above corollary; we obtain the following result:

**Corollary 7.** *If $0 < a < \frac{\sqrt{6}}{\pi}$, then the function $\mathcal{E}_{2,2}^{(a)}(s;z)$ is uniformly convex in $\mathcal{D}$.*

**Theorem 7.** *Assume that of all the hypotheses of Theorem 5 hold true. Then the function $\mathcal{E}_{\kappa,\nu}^{(a)}(s;z)$ is strongly convex in $\mathcal{D}$.*

**Proof.** Let $z \in \mathcal{D}$; a short computation gives

$$\left| \left[ z(\mathcal{E}_{\kappa,\nu}^{(a)}(s;z))' \right]' - 1 \right| \leq \sum_{n=1}^{\infty} \frac{(n+1)^2 a^s\Gamma(\nu)}{(n+a)^s\Gamma(n\kappa + \nu)}$$

$$=: a^s\Gamma(\nu) \sum_{n=1}^{\infty} \frac{f_n(\nu, \kappa)}{(n+a)^s}. \qquad (36)$$

Let us write the expression of the sequence $(f_n(\nu, \kappa))_{n\geq 1}$ as follows:

$$f_n(\nu, \kappa) = \frac{\xi_n}{\Gamma(n+1)} \cdot \left( \frac{n+1}{n} \right) \quad (n \geq 1).$$

Then the sequence $f_n(v, \kappa)$ is decreasing as the product of three decreasing sequences. Thus, we get

$$
\begin{aligned}
\left| \left[ z(\mathcal{E}_{\kappa,v}^{(a)}(s;z))' \right]' - 1 \right| &\leq \frac{4a^s \Gamma(v)}{\Gamma(\kappa + v)} \sum_{n=1}^{\infty} \frac{1}{(n+a)^s} \\
&\leq \frac{4a^s \Gamma(v)}{\Gamma(\kappa + v)} \sum_{n=1}^{\infty} \frac{1}{n^s} \\
&= \frac{4a^s \Gamma(v) \zeta(s)}{\Gamma(\kappa + v)} = \eta.
\end{aligned} \tag{37}
$$

Thus, we get

$$
\left[ z(\mathcal{E}_{\kappa,v}^{(a)}(s;z))' \right]' \prec 1 + \eta z \quad (\forall z \in \mathcal{D}),
$$

and consequently, we have

$$
\left| \arg \left\{ \left[ z(\mathcal{E}_{\kappa,v}^{(a)}(s;z))' \right]' \right\} \right| < \arcsin(\eta) \quad (\forall z \in \mathcal{D}). \tag{38}
$$

By applying Lemma 3 once more, where $m = 0$ and the functions $F$ and $G$ are defined by

$$
F(z) = 1 + \eta z \quad \text{and} \quad G(z) = \left[ z(\mathcal{E}_{\kappa,v}^{(a)}(s;z))' \right]' \quad (z \in \mathcal{D}),
$$

we obtain

$$
(\mathcal{E}_{\kappa,v}^{(a)}(s;z))' \prec 1 + \frac{\eta}{2} z \quad (\forall z \in \mathcal{D}).
$$

Having the above inequality and (38) in mind, then for $z \in \mathcal{D}$, we have

$$
\begin{aligned}
\left| \arg \left\{ \frac{(z(\mathcal{E}_{\kappa,v}^{(a)}(s;z))')'}{(\mathcal{E}_{\kappa,v}^{(a)}(s;z))'} \right\} \right| &= \left| \arg \left( (z(\mathcal{E}_{\kappa,v}^{(a)}(s;z))')' \right) - \arg \left( (\mathcal{E}_{\kappa,v}^{(a)}(s;z))' \right) \right| \\
&\leq \left| \arg \left( (z(\mathcal{E}_{\kappa,v}^{(a)}(s;z))')' \right) \right| + \left| \arg \left( (\mathcal{E}_{\kappa,v}^{(a)}(s;z))' \right) \right| \\
&\leq \arcsin(\eta) + \arcsin \left( \frac{\eta}{2} \right) \\
&= \arcsin \left( \frac{\eta}{2} \left[ \sqrt{4 - (\eta)^2} + \sqrt{1 - (\eta)^2} \right] \right) \\
&= \frac{\vartheta \pi}{2}.
\end{aligned}
$$

So the proof of Theorem 7 is completed. $\square$

**Theorem 8.** *Let the parameter ranges for $\kappa, v, a > 0$, and $s \geq 0$ be such that $2a^s \Gamma(v) \leq (a+1)^s \Gamma(v + \kappa)$. If the following inequality is valid:*

$$
\kappa \log(v) - \frac{\kappa}{v} > 1, \tag{39}
$$

*then the function $\mathcal{E}_{\kappa,v}^{(a)}(s;z)$ is close-to-convex with respect to the function $-\log(1 - z)$.*

**Proof.** Firstly, we note that the condition

$$
\frac{\Gamma(v)}{\Gamma(v + \kappa)} \leq \frac{(a+1)^s}{2a^s}
$$

implies $\rho_1^{(a)}(\nu, \kappa, s) \geq 2\rho_2^{(a)}(\nu, \kappa, s)$. In addition, for $n \geq 2$, we find that

$$n\rho_n^{(a)}(\nu, \kappa, s) - (n+1)\rho_{n+1}^{(a)}(\nu, \kappa, s) \geq \frac{a^s \Gamma(\nu)}{(n+a)^s} \left[ \frac{n-1}{\Gamma(n\kappa + \nu - \kappa)} - \frac{n}{\Gamma(n\kappa + \nu)} \right] \quad (40)$$

However, in [30], Proof of Theorem 4.1, it was proved that the sequence $(\psi_n)_{n\geq 2}$ defined by

$$\psi_n = \frac{n-1}{\Gamma(n\kappa + \nu - \kappa)} \quad (n \geq 2),$$

is decreasing if (39) is valid. From this observation and with the aid of (40), we deduce that the sequence $(n\rho_n^{(a)}(\nu, \kappa, s))_{n\geq 2}$ is decreasing. Thus, finally, by applying Lemma 4, we readily establish the desired result asserted by Theorem 8. $\quad \square$

Specifying $(\kappa, \nu) = (2, 3)$ in Theorem 8, we compute the following corollary:

**Corollary 8.** *For all $a, s > 0$, the function $\mathcal{E}_{2,3}^{(a)}(s; z)$ is close-to-convex with respect to the function $-\log(1-z)$.*

**Theorem 9.** *Let the parameter space be the same as in Theorem 8. Then, we get*

$$\Re\left( \frac{\mathcal{E}_{\kappa,\nu}^{(a)}(s; z)}{z} \right) > \frac{1}{2} \quad (\forall z \in \mathcal{D}).$$

**Proof.** From (8), we have

$$\frac{\mathcal{E}_{\kappa,\nu}^{(a)}(s; z)}{z} = \sum_{n=1}^{\infty} \rho_n^{(a)}(\nu, \kappa, s) z^{n-1}.$$

Under Condition (39), the sequence $(\psi_n)_{n\geq 2}$ is decreasing, and consequently, the sequence $\{\psi_n / (n-1)\}_{n\geq 2}$ is also decreasing. Therefore, the sequence $\left\{ \rho_n^{(a)}(\nu, \kappa, s) \right\}_{n\geq 2}$ is decreasing. In addition, the inequality $(a+1)^s \Gamma(\nu + \kappa) \geq a^s \Gamma(\nu)$ implies $\rho_1^{(a)}(\nu, \kappa, s) \geq \rho_2^{(a)}(\nu, \kappa, s)$. Next, we show that the sequence $\left\{ \rho_n^{(a)}(\nu, \kappa, s) - \rho_{n+1}^{(a)}(\nu, \kappa, s) \right\}_{n\geq 1}$ is also decreasing. So it suffices to prove that

$$\rho_n^{(a)}(\nu, \kappa, s) - 2\rho_{n+1}^{(a)}(\nu, \kappa, s) \geq 0 \quad (\forall n \geq 1).$$

Then, we get

$$\rho_n^{(a)}(\nu, \kappa, s) - 2\rho_{n+1}^{(a)}(\nu, \kappa, s) \geq \frac{a^s \Gamma(\nu)}{(n+a)^s} \left[ \frac{1}{\Gamma(n\kappa + \nu - \kappa)} - \frac{2}{\Gamma(n\kappa + \nu)} \right]. \quad (41)$$

Since the function $z \mapsto \frac{\Gamma(z+\delta)}{\Gamma(z)}$ is increasing on $(0, \infty)$ for each $\delta > 0$, we have

$$\frac{\Gamma(n\kappa + \nu)}{\Gamma(n\kappa + \nu - \kappa)} \geq \frac{\Gamma(\kappa + \nu)}{\Gamma(\nu)}. \quad (42)$$

We set $s = 0$ in our assumption; we obtain

$$\Gamma(\kappa + \nu) \geq 2\Gamma(\nu),$$

and in view of the above inequality and (41), we deduce that

$$\rho_n^{(a)}(\nu, \kappa, s) - 2\rho_{n+1}^{(a)}(\nu, \kappa, s) \geq 0 \quad (\forall n \geq 1).$$

Hence, the sequence $\left\{\rho_n^{(a)}(\nu,\kappa,s) - \rho_{n+1}^{(a)}(\nu,\kappa,s)\right\}_{n\geq 1}$ is decreasing. So Lemma 5 yields the asserted result. The proof of Theorem 9 is completed. □

## 3. Conclusions

In the present paper, we have considered a class of functions related to the Barnes–Mittag-Leffler function (consult (8)). We found sufficient conditions to be imposed on the parameters of the aforementioned function that allow us to conclude that it possesses certain geometric properties such as starlikenes, uniformly starlike (convex), strongly starlike (convex), convexity, and close-to-convexity in the unit disk. Further, in view of Remarks 1–3, we deduce that some of the results derived in the present paper improved the corresponding results established in the literature.

Hopefully, the mathematical tools in the proof of the original results contained in our present paper will stimulate researchers' imaginations and inspire them to also study some other special functions such as the Fox–Wright function and the generalized Hurwitz–Lerch-type functions.

**Author Contributions:** Writing—review & editing, A.A. and K.M. All authors have read and agreed to the published version of the manuscript.

**Funding:** The authors extend their appreciation to the Deanship of Scientific Research at Northern Border University, Arar, KSA for funding this research work through the project number "NBU-FFR-2023-0159".

**Data Availability Statement:** The data presented in this study are available on request from the corresponding author. The data are not publicly available due to privacy.

**Conflicts of Interest:** The authors declare no conflicts of interest.

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
