# Peer review of "Further Geometric Properties of the Barnes–Mittag-Leffler Function"

_axioms, doi:10.3390/axioms13010012_

Round 1

Reviewer 1 Report

Comments and Suggestions for Authors

I think this paper is interesting and the main idea is novel, Results are correct and new However, there are still some small issues worth further improvement and discussion. See the attached file

Comments on the Quality of English Language

Minor editing required

Author Response

Thank you very much  for your comments about the present paper.

Reviewer 2 Report

Comments and Suggestions for Authors

see pdf file

Author Response

Thank you very much for your comments. Please see corrections request in the revised version (in blue color).

Reviewer 3 Report

Comments and Suggestions for Authors

Manuscript “Further geometric properties of the Barnes Mittag-Leffler function” can be published in Axioms since it consists of sufficient conditions for certain geometric properties the Barnes Mittag-Leffler function such as uniformly starlike (convex), strongly starlike (convex), convexity, close-to-convexity in the unit disk. There are several minor remarks:

1.      Excessive citing of own publications.

2.      There are two different definitions for ρ˜^(a)_{n+1}(ν, κ, s), one given after equation (10), and another given in equation (15).  

Author Response

Thank you very much for your comments:

  1.  I reduced the number of citations to my own publications.
  2.  I correct it the notations of definitions after equation (10) and in equation (15). 

Reviewer 4 Report

Comments and Suggestions for Authors

  This paper is a continuation of the published work of the authors in recent years.  Sufficient conditions are provided on the parameters of a class of functions related to the Barnes-Mittag-Leiffler function which allow us to conclude that it is possesses certain  geometric properties in the unit disk. They use as a key tools in their demonstrations the monotonicity properties of certain class of functions related to the Gamma function.

   The proofs are clear presented but some more examples and applications could be given. In addition, the conclusions section could be improved. The references are relevant and the introduction is good written.

     I recommend this paper for publication in this journal.

Author Response

Thank you very much for your comments about our paper.

Reviewer 5 Report

Comments and Suggestions for Authors

The peer reviewed manuscript considers the properties of the Barnes-Mittag-Leffler (BML) functions. Currently, certain classes of Mittag-Leffler-type functions are increasingly used in fractional calculus and its applications in modern problems of mathematical physics, which motivates the relevance of the research.

The authors continue their line of research in the field of special functions and focus on the geometric properties of the BML functions such as starlikness, uniformly starlike/convex, strongly starlike/ convex, convexity, close-to-convexity for normalized form of the BML functions. The normalization condition is introduced in the work.

Section 1 introduces the necessary definitions and lemmas.

The main results of the work are formulated in Theorems 1-9. For example, the properties that  the BML function is starlike (convex) of are established in Theorems 1 and 2, respectively, under certain conditions given in these theorems. Similar results on the properties of the normalized BML functions are considered in the remaining theorems.

Proofs are given for all main theorems.

In conclusion, the authors discuss possible consequences of the obtained results in the theory of special functions and further prospects.

Two minor comments are given below.

1. In the definition of starlike function on p. 2, line 4 at the top, the designation of the symbol \mathfrak{R} in the inequality should be clarified for greater readability.

2. The list of references is somewhat overloaded with self-citation; in my opinion, not all references from [6-8] are directly related to the work.

At the same time, some works related to this study remained outside the attention of the authors, for example, A.Lecko, A. Wisniowska, J.Comp.App.Math.2003,155,383.

In general, the results obtained in this work contribute to the theory of special functions and its possible applications.

The work is written clearly and well structured. The reference list includes all necessary major publications in the field of study.

A positive aspect of the work is a possible prospects for further research based on the results of the study.

Summarizing, I believe that the peer-reviewed paper work can be accepted for publication in Axioms after eliminating the minor comments made.

Author Response

Thank you very much for your kind remarks.

I reduced the number of citations associated with references and deleted the references [6–8] in the first version.